# Dynamic evolution of utilization efficiency of medical and health services in China

**Jing Zhang** [ID] *

Medical Economic and Management School of Anhui University of Chinese Medicine, Hefei, China

* zhang0216@ahtcm.edu.cn

## Abstract

In order to optimize the Chinese medical and health system and improve people's health level, the SFA Malmquist model, the spatial econometric model, and the standard deviation ellipse method were used to measure the efficiency of medical and health services in China's 31 provinces between 2010 and 2020. Study results indicated that the average efficiency value of the 31 provinces generally exceeded 0.8. Specifically, the average efficiency values in the eastern and central regions increased from 0.852 to 0.875 and from 0.858 to 0.88, respectively. In the western and northeastern regions, these values rose from 0.804 to 0.835 and from 0.827 to 0.854, respectively. From the perspective of spatial distribution, there were high-high and low-low clusters in most provinces with significant spatial dependence among them. This analysis reveals that medical and health services efficiency in China demonstrates a spatial pattern extending from northeast to southwest.

## 1. Introduction

As one of the most crucial basic public services, medical and health services (MHS) represent a vital aspect of social fairness and justice. Their efficiency is directly related to people's lives, health, and the stable development of the social economy [1]. MHS has emerged as a focal topic within academia and industry. Consequently, the key to achieving healthy social development lies in improving and optimizing MHS efficiency [2]. The Chinese government has exerted considerable efforts in this regard. Since the healthcare system reform in 2009, the healthcare industry has witnessed significant progress, with notable improvements in the quality and quantity of MHS [3, 4]. The number of medical and health organizations reached 1.03095 million, the average life expectancy of the population increased from 67.77 in 1981 to 78.2, and the maternal mortality rate declined from 80 deaths per 100,000 pregnant women in 1991 to 16 deaths per 100,000 at the end of December 2021, according to data from the National Bureau of Statistics [5]. These advancements have alleviated the substantial pressure on the healthcare system and reflect the optimization of healthcare resource allocation and the further improvement and significant progress of healthcare services and resource utilization efficiency. However, the Chinese MHS industry has also faced numerous challenges. According to economic and health data from 2012 to 2017, the proportion of healthcare expenses to the country's GDP increased from 4.8% to 6.1%, remaining below that of most Organization for Economic Co-operation and Development (OECD) member countries during the same period [6]. Moreover, the productivity of medical services in China was markedly low; the

**Data Availability Statement:** Due to the fact that the original data is internal materials of the project, it cannot be publicly shared, and all data access needs to be approved by Anhui University of Traditional Chinese Medicine before the project is officially concluded. Researchers who meet the

standards for accessing confidential data can obtain data from the Ethics Committee of Anhui University of Chinese Medicine (through yyjjglxy@sina.com contact).

**Funding:** The author(s) received no specific funding for this work.

**Competing interests:** The authors declare that there is no conflict of interest.

annual average growth rate (AAGR) of Chinese total healthcare expenditure was 11.28% from 2010 to 2021 [7]. As the core indicator of medical service output, the overall annual growth rate of outpatients was only 3.15% [8]. As the core indicator of medical service investment, the AAGR of health technicians and hospitals was only 5.56% and 4.76%, respectively [9]. Therefore, this study aims to explore the changes in the efficiency of Chinese MHS since the reform and to assess whether there is an imbalance in the development of MHS efficiency among different provinces. The answers to these questions are essential for the healthy development of the current Chinese healthcare system.

Some scholars have conducted extensive research on medical services efficiency. Li Yi (2021) [10] explored MHS efficiency and its interfering factors based on medical and health data from 30 Chinese provinces from 2005 to 2018. The results indicated that MHS efficiency demonstrated a strong trend in the East but was weaker in the West and Central. The interfering factors included the developing level of regional economy, technology education, regional aging, and transportation levels. Zhang Mei (2022) [11] evaluated the public MHS efficiency in traditional Chinese medicine hospitals from 2012 to 2020 and found that overall medical services efficiency was at a moderate level with significant regional differences. The external environment, like the average times and sick beds per day for which Chinese medicine hospital physicians were responsible for treatment and diagnosis, significantly affected the MHS efficiency. Luan Yunyun (2022) [12] analyzed the interfering factors of MHS efficiency in China and found that population density, government health budget expenditures, and the non-market level of medical services significantly influenced medical services efficiency. These studies suggest that medical services efficiency is not only related to the type of medical institution and organizational management but is also significantly influenced by exogenous variables such as population density.

In terms of efficiency research methods, published research has primarily focused on Data Envelopment Analysis (DEA). For example, Hao Jinwei (2020) [13] employed the Bootstrap DEA method to calculate the operational medical services efficiency in 22 town health centers in Hubei Province from 2014 to 2016. The research showed that the utilization rate of health service resources by town health centers was not high. Yu Benhai (2024) [14] utilized the three-stage DEA model to evaluate service efficiency in primary healthcare institutions in China. Zhang Xiaoxi (2021) [15] applied DEA to study the MHS efficiency of the Yangtze River Delta regions between 2010 and 2019 and discovered that the efficiency characteristics of medical services varied across different regions. International literature also focuses on efficiency research; similarly, scholars have used Data Envelopment Analysis and the Malmquist approach to estimate efficiency. Some studies employed meta-frontier analysis, but unlike Chinese studies, the objectives of the foreign research primarily involved energy efficiency, and included regions such as G7 economies, G20 countries, South Asian countries, and Chinese provinces in different periods [16–24]. However, DEA does not account for the impact of statistical errors. Moreover, when evaluating influencing efficiency components, a two-step analysis combining DEA and regression can produce inaccurate estimation results [25]. Compared with the DEA model, the Stochastic Frontier Analysis (SFA) is a mixed-effects model and a specific case of a one-step estimation method, which can ensure the reliability of efficiency calculation results and the accuracy of impact factor analysis. Thus, this article has chosen SFA as the main method for evaluating efficiency.

This article introduces innovations in expanding research contents and applying research methods. Although the current literature on the analysis and evaluation of MHS efficiency is comparatively rich, providing a basic and important foundation for related research, it still exhibits several shortcomings. On one hand, previous studies have primarily focused on efficiency evaluation and analysis, with a lack of analysis on the spatial dependence and

agglomeration of MHS efficiency. On the other hand, unlike the commonly used DEA method for evaluating efficiency, this study, taking China's 31 provinces as examples and using panel data from 2010 to 2020, employs the Stochastic Frontier Approach (SFA) model, the Malmquist index, the Standard Deviation Ellipse, and the spatial econometric model to explore the spatial dependence and dynamic evolution characteristics of MHS efficiency, thereby deepening the research on the dynamic changes in medical services efficiency.

The contributions of this study include revealing the efficiency changes in the Chinese medical service system since the "China New Medical Reform," clarifying the differences among various provinces, providing a practical basis for the subsequent improvement of medical and health service efficiency, and offering the corresponding paths and directions for medical institution reform.

# 2. Research methods and data

## 2.1 Research methods

**2.1.1. Stochastic frontier approach.** Certainty models fail to consider influence factors that are both random and beyond the control of producers. Therefore, this article employs the stochastic frontier model of production to analyze these elements. Under a given technological input, the production function is interpreted as the maximum potential yield. The foundational model is outlined as follows:

$$y_i = f(x_i; \beta) \bullet \exp(-\mu_i), i = 1, 2, 3 \cdots, n \tag{1}$$

In Eq (1), $y_i$ is the input vector of the $i$ unit, $x_i$ is the parameter vector used for thumb suck, $f$ represents the production function, $\mu_i$ represents the inefficiency of the i unit which is a non negative number.

Scholars such as Meeusen and Broeck (1977) [26] initially introduced the SFA method, which divides random disturbances into two components. The first component consists of statistical errors induced by stochastic errors $v_i$, such as measurement errors, statistical errors, and other factors beyond the control of decision units. The second component is the technical inefficiency term $\mu_i$, which represents factors leading to low efficiency. They highlighted that the output in the production process is influenced by uncontrollable non-human factors including stochastic effects, natural disasters, geography, and climate. Therefore, the forefront production function for a decision-making unit is stochastic rather than deterministic. From Eq (1), the stochastic forefront production function is represented:

$$y_i = f(x_i; \beta) \bullet \exp\{v_i\} \exp(-\mu_i), i = 1, 2, 3 \cdots, n \tag{2}$$

If we take the Cobb Douglas production function, we can obtain the model:

$$Y = AL^\alpha K^\beta e^{1-\mu} \tag{3}$$

By having logarithmic linearization on both sides, new equation can be obtained:

$$\ln y_i = \beta_0 + \sum_{i=1}^{n} \beta_i \ln x_i + v_i - \mu_i, i = 1, 2, 3 \cdots n \tag{4}$$

Among them, $v_i$ and $\mu_i$ are independent and not related to explanatory variables, ultimately, the technical efficiency (TE) is written as:

$$TE_i = \frac{y_i}{f(x_i; \beta) * \exp(v_i)} = \exp\{-\mu_i\}, i = 1, 2, 3 \cdots n \tag{5}$$

**2.1.2. Malmquist index.** Due to the limitations of the SFA model in only providing static evaluations of the efficiency of each decision-making unit, it fails to reveal the dynamic evolution of efficiency or the underlying factors influencing efficiency changes. Therefore, the Malmquist index is utilized to analyze the total factor productivity (TFP) of medical and health services in China. The Malmquist index, an extension of the DEA method, is a specialized index analysis method for measuring the growth and change of TFP. It allows for the analysis of the efficiency evolution of decision-making units across different periods and can decompose the changes in total factor productivity (tfpch) into technology changes (techch) and efficiency changes (effch) [27]. This method helps avoid attributing changes in decision-making unit efficiency to a single index while neglecting the influence of another, thereby providing a more detailed understanding of the sources of comprehensive efficiency improvement. This method is widely used in the calculation of production efficiency in sectors such as finance, industry, and healthcare, where comparative research is based on the results of efficiency calculations. Effch represents the degree of progress brought about by innovation between two periods, while techch measures the impact of changes at the forefront of production. Further, Effch can be divided into scale efficiency (sech) and pure efficiency (pech). Scale efficiency (sech) evaluates the appropriateness of a decision-making unit's (DMU) scale, whereas pure efficiency (pech) assesses management efficiency, which can be decomposed into technology (effch), scale efficiency (sech), technological progress (techch), and pure technological efficiency (pech) [28].

Assuming the output of period t is $(x^t, y^t)$, the output of period t+1 is $(x^{t+1}, y^{t+1})$, then the output distance functions of period t and period t+1 are $D^t(x^t, y^t)$, $D^{t+1}(x^{t+1}, y^{t+1})$ respectively, representing the distance between the production configuration and the current production frontier. The Malmquist index can be showed as [29]:

$$Malmquist(TFP) = M_{t,t+1}(x^t, y^t, x^{t+1}, y^{t+1}) = \frac{D^{t+1}(x^{t+1}, y^{t+1})}{D^t(x^t, y^t)} \times \sqrt{\left[\frac{D^t(x^{t+1}, y^{t+1})}{D^{t+1}(x^{t+1}, y^{t+1})} \times \frac{D^t(x^t, y^t)}{D^{t+1}(x^t, y^t)}\right]} \quad (6)$$

In Eq (6), the Malmquist index has three possible outcomes: if it is greater than 1, it indicates an increase in total factor productivity; if it is less than 1 [30], it signifies a decrease in total factor productivity; if it is exactly 1, it denotes that the total factor productivity has remained unchanged.

**2.1.3. Standard deviation ellipse.** The Standard Deviation Ellipse is used to analyze the directional features of spatial distribution, which primarily involves spatial basic parameters such as the center, long axis standard deviation, short axis standard deviation, and azimuth of ellipses. These parameters quantitatively depict the spatial distribution features of economic attributes [31]. The basic parameters are expressed as follows:

$$\text{Center}: \qquad \overline{x_w} = \sum_{i=1}^{n} w_i x_i / \sum_{i=1}^{n} w_i \cdots \overline{Y_w} = \sum_{i=1}^{n} w_i y_i / \sum_{i=1}^{n} w_i \qquad (7)$$

$$\text{Long axis standard deviation}: \qquad \sigma_x = \sqrt{\left(\sum_{i=1}^{n} w_i \tilde{x} \text{os}\theta - w_i \tilde{y}\sin\theta\right)^2 / \sum_{i=1}^{n} w_i^2} \quad (8)$$

$$\text{Short axis standard deviation}: \qquad \sigma_y = \sqrt{\left(\sum_{i=1}^{n} w_i \tilde{x}_i \cos\theta - w_i \tilde{y}_i\sin\theta\right)^2 / \sum_{i=1}^{n} w_i^2} \quad (9)$$

$$\text{Azimuth}: \quad \tan(\theta) = \left[ (\sum_{i=1}^{n} \tilde{x}_i^2 - \sum_{i=1}^{n} \tilde{y}_i^2) + \left( \sqrt{\left( \sum_{i=1}^{n} \tilde{x}_i^2 - \sum_{i=1}^{n} \tilde{y}_i^2 \right)^2 + 4 \left( \sum_{i=1}^{n} \tilde{x}_i \tilde{y}_i \right)^2} \right) / 2 \sum_{i=1}^{n} \tilde{x}_i \tilde{y}_i \right] \quad (10)$$

In Eqs (7)–(10), $x_i$ and $y_i$ are the longitude and latitude coordinates of the geographical location center of the study object; $\tilde{x}$ and $\tilde{y}$ represent the coordinates of the distance from each point to regional center of gravity respectively; $w_i$ is weight; $N$ represents the number of research subjects; The azimuth of an ellipse $\theta$ is the angle formed by the main axis of the ellipse in a clockwise direction from north [32, 33].

**2.1.4. Spatial econometric model.** The spatial correlation test employs Moran's I index, introduced by Moran in 1950, to assess whether variables exhibit regional correlation and spatial dependence. Moran's I index is defined as follows [34]:

$$Moran's\ I = \frac{\sum_{i=1}^{n} \sum_{j=1}^{n} w_{ij}(y_i - \overline{y})(y_j - \overline{y})}{s^2 \sum_{i=1}^{n} \sum_{j=1}^{n} w_{ij}}, s^2 = 1/2 * \sum_{i=1}^{n} (y_i - \overline{y})^2, \overline{y} = 1/n \sum_{i=1}^{n} y_i \quad (11)$$

In Eq (11), $y_i$ is the efficiency index value of medical services for the $i$ province, and $n$ is the number of provinces; $w_{ij}$ is geographic inverse distance spatial weight matrix [35, 36]. Typically, the range of Moran's I index values is [−1, 1]. If there is positive spatial interdependence (Moran's I>0), it suggests that provinces with higher (or lower) levels of MHS efficiency exhibit higher spatial clustering. Conversely, if there is negative spatial interdependence (Moran's I<0), it indicates spatial disparities in MHS efficiency between a province and its neighbors. If the index is zero (Moran's I = 0), MHS efficiency in each province is randomly distributed in space, suggesting no spatial dependence.

## 2.2 Selection of SFA indicators and data sources

**2.2.1. Investment indicators.** The number of beds, per capita net assets, and the number of technical personnel of medical facilities are selected as input indices. These indices respectively reflect the scale and carrying capacity of medical facilities, the investment status of resources and medical security strength, and the service level and capacity of medical facilities [37–39].

**2.2.2. Output index.** The income, number of diagnoses, and discharges of medical institutions are selected as output indices, reflecting their business performance, workload and work efficiency, and service processing capacity. To address the bias of these three selected indices and to maximize the information contained within the output indices, the logarithm of the three indicators was taken. The output indices were then weighted and synthesized through principal component analysis. Subsequently, the output index was used as the dependent variable in the stochastic frontier production function model [40–42].

**2.2.3. Data source.** The input and output data of medical resources across the 31 provinces of China were sourced from the 2010–2020 editions of the "China Statistical Yearbook," "China Health Statistical Yearbook," and "China Health and Family Planning Statistical Yearbook" [43]. Ethical issues (Including plagiarism, informed consent, misconduct, data fabrication and/or falsification, double publication and/or submission, redundancy, etc.) have been completely observed by the authors. All methods were carried out in accordance with relevant guidelines and regulations. The research project has been supervised and approved by the

Ethics Committee of Anhui University of Chinese Medicine. All data does not involve human participants.

# 3. Spatiotemporal evolution of medical and health services efficiency

## 3.1 Time evolution of medical and health services efficiency

**3.1.1. Static analysis based on SFA model.** Using Frontier 4.1 software, the MHS efficiency of China's 31 provinces from 2010 to 2020 was evaluated using the SFA model. The calculation outcomes are presented in Table 1.

**Table 1. Efficiency values of inter provincial MHS between 2010 and 2020.**

| region\year | 2010 | 2011 | 2012 | 2013 | 2014 | 2015 | 2016 | 2017 | 2018 | 2019 | 2020 | Average value |
|---|---|---|---|---|---|---|---|---|---|---|---|---|
| **Beijing** | 0.834 | 0.836 | 0.839 | 0.842 | 0.845 | 0.847 | 0.850 | 0.852 | 0.855 | 0.857 | 0.860 | 0.847 |
| **Tianjin** | 0.800 | 0.803 | 0.807 | 0.810 | 0.813 | 0.816 | 0.819 | 0.822 | 0.825 | 0.828 | 0.831 | 0.816 |
| **Hebei** | 0.875 | 0.877 | 0.879 | 0.881 | 0.883 | 0.885 | 0.887 | 0.889 | 0.891 | 0.893 | 0.894 | 0.885 |
| **Shanghai** | 0.843 | 0.846 | 0.848 | 0.851 | 0.853 | 0.856 | 0.858 | 0.861 | 0.863 | 0.865 | 0.868 | 0.856 |
| **Jiangsu** | 0.891 | 0.893 | 0.895 | 0.896 | 0.898 | 0.900 | 0.902 | 0.903 | 0.905 | 0.906 | 0.908 | 0.900 |
| **Zhejiang** | 0.881 | 0.883 | 0.885 | 0.887 | 0.889 | 0.891 | 0.893 | 0.895 | 0.896 | 0.898 | 0.900 | 0.891 |
| **Fujian** | 0.842 | 0.844 | 0.847 | 0.850 | 0.852 | 0.855 | 0.857 | 0.860 | 0.862 | 0.864 | 0.866 | 0.854 |
| **Shandong** | 0.896 | 0.898 | 0.899 | 0.901 | 0.903 | 0.904 | 0.906 | 0.907 | 0.909 | 0.911 | 0.912 | 0.904 |
| **Guangdong** | 0.908 | 0.909 | 0.911 | 0.912 | 0.914 | 0.915 | 0.917 | 0.918 | 0.919 | 0.921 | 0.922 | 0.915 |
| **Hainan** | 0.755 | 0.759 | 0.763 | 0.767 | 0.771 | 0.775 | 0.779 | 0.782 | 0.786 | 0.789 | 0.793 | 0.774 |
| **Eastern region** | 0.852 | 0.855 | 0.857 | 0.860 | 0.862 | 0.864 | 0.867 | 0.869 | 0.871 | 0.873 | 0.875 | 0.864 |
| **Shanxi** | 0.819 | 0.822 | 0.825 | 0.828 | 0.831 | 0.834 | 0.836 | 0.839 | 0.842 | 0.845 | 0.847 | 0.870 |
| **Anhui** | 0.861 | 0.863 | 0.865 | 0.868 | 0.870 | 0.872 | 0.874 | 0.876 | 0.878 | 0.881 | 0.883 | 0.871 |
| **Jiangxi** | 0.843 | 0.846 | 0.849 | 0.851 | 0.854 | 0.856 | 0.859 | 0.861 | 0.863 | 0.866 | 0.868 | 0.868 |
| **Henan** | 0.891 | 0.892 | 0.894 | 0.896 | 0.898 | 0.900 | 0.901 | 0.903 | 0.904 | 0.906 | 0.908 | 0.866 |
| **Hubei** | 0.867 | 0.869 | 0.871 | 0.873 | 0.875 | 0.877 | 0.879 | 0.881 | 0.883 | 0.885 | 0.887 | 0.867 |
| **Hunan** | 0.865 | 0.867 | 0.870 | 0.872 | 0.874 | 0.876 | 0.878 | 0.880 | 0.882 | 0.884 | 0.886 | 0.863 |
| **Central region** | 0.858 | 0.860 | 0.862 | 0.865 | 0.867 | 0.869 | 0.871 | 0.873 | 0.876 | 0.878 | 0.880 | 0.867 |
| **Neimenggu** | 0.804 | 0.807 | 0.811 | 0.814 | 0.817 | 0.820 | 0.823 | 0.826 | 0.829 | 0.832 | 0.835 | 0.820 |
| **Chongqing** | 0.828 | 0.831 | 0.834 | 0.836 | 0.839 | 0.842 | 0.845 | 0.847 | 0.850 | 0.852 | 0.855 | 0.842 |
| **Sichuan** | 0.885 | 0.887 | 0.889 | 0.890 | 0.892 | 0.894 | 0.896 | 0.898 | 0.899 | 0.901 | 0.903 | 0.894 |
| **Guizhou** | 0.829 | 0.832 | 0.834 | 0.837 | 0.840 | 0.843 | 0.845 | 0.848 | 0.850 | 0.853 | 0.855 | 0.842 |
| **Yunnan** | 0.850 | 0.852 | 0.855 | 0.857 | 0.860 | 0.862 | 0.864 | 0.866 | 0.869 | 0.871 | 0.873 | 0.862 |
| **Tibet** | 0.676 | 0.682 | 0.687 | 0.692 | 0.698 | 0.703 | 0.708 | 0.713 | 0.717 | 0.722 | 0.727 | 0.702 |
| **Shaanxi** | 0.836 | 0.838 | 0.841 | 0.844 | 0.846 | 0.849 | 0.851 | 0.854 | 0.856 | 0.859 | 0.861 | 0.849 |
| **Gangsu** | 0.808 | 0.812 | 0.815 | 0.818 | 0.821 | 0.824 | 0.827 | 0.830 | 0.833 | 0.835 | 0.838 | 0.824 |
| **Qinghai** | 0.728 | 0.732 | 0.737 | 0.741 | 0.745 | 0.750 | 0.754 | 0.758 | 0.762 | 0.766 | 0.770 | 0.749 |
| **Ningxia** | 0.744 | 0.749 | 0.753 | 0.757 | 0.761 | 0.765 | 0.769 | 0.773 | 0.777 | 0.781 | 0.784 | 0.765 |
| **Xinjiang** | 0.813 | 0.816 | 0.819 | 0.822 | 0.825 | 0.828 | 0.831 | 0.834 | 0.836 | 0.839 | 0.842 | 0.828 |
| **Guangxi** | 0.849 | 0.852 | 0.854 | 0.857 | 0.859 | 0.861 | 0.864 | 0.866 | 0.868 | 0.871 | 0.873 | 0.861 |
| **Western Region** | 0.804 | 0.807 | 0.811 | 0.814 | 0.817 | 0.820 | 0.823 | 0.826 | 0.829 | 0.832 | 0.835 | 0.820 |
| **Liaoning** | 0.845 | 0.847 | 0.850 | 0.852 | 0.855 | 0.857 | 0.860 | 0.862 | 0.864 | 0.867 | 0.869 | 0.857 |
| **Jilin** | 0.812 | 0.816 | 0.819 | 0.822 | 0.825 | 0.828 | 0.831 | 0.833 | 0.836 | 0.839 | 0.842 | 0.827 |
| **Heilongjiang** | 0.825 | 0.828 | 0.831 | 0.834 | 0.836 | 0.839 | 0.842 | 0.844 | 0.847 | 0.850 | 0.852 | 0.839 |
| **Northeast region** | 0.827 | 0.830 | 0.833 | 0.836 | 0.839 | 0.841 | 0.844 | 0.847 | 0.849 | 0.852 | 0.854 | 0.841 |
| **Nationwide** | 0.832 | 0.835 | 0.838 | 0.841 | 0.843 | 0.846 | 0.849 | 0.851 | 0.854 | 0.856 | 0.858 | 0.845 |

Overall, from 2010 to 2020, the average MHS efficiency in China varied between 0.832 and 0.858, indicating that the overall efficiency was moderate but still had significant potential for further improvement. Temporally, the average MHS efficiency in China increased from 0.832 in 2010 to 0.858 in 2020, demonstrating a steady growth trend with the output of Chinese MHS efficiency progressively approaching the forefront of production. Regionally, MHS efficiency displayed differentiated characteristics where "the eastern and central regions were high, but the northwestern regions were low." Specifically, the average change range of MHS efficiency in the eastern regions from 2010 to 2020 was 0.852 to 0.875, in the central regions was 0.858 to 0.88, in the northeastern regions was 0.827 to 0.854, and in the western regions was 0.804 to 0.835. The average MHS efficiency in the eastern and central regions was slightly higher than the national average (0.845), with the eastern areas consistently leading and becoming the "main contributors" to national efficiency. However, the northeast and western areas lagged behind the national average (0.845). In terms of growth rates, the average annual increase rates of MHS efficiency in the eastern, central, western, and northeastern regions were 2.55%, 2.8%, 3.87%, and 3.26% respectively. Although MHS efficiency in the western and northeastern regions was slightly lower, their growth rates were relatively high. From a regional perspective, there were considerable differences in MHS efficiency among provinces, but these internal regional differences could not be fully captured by regional classifications alone. For example, in the eastern regions, the average MHS efficiency in Jiangsu was as high as 0.9, while in Hainan it was only 0.774, lower than the national average. In the western regions, the average MHS efficiency in Sichuan was 0.894, while that in Tibet was only 0.702. To further explore the areal differences in MHS efficiency in China, K-means clustering analysis was employed using SPSS 21.0 software to reclassify the average MHS efficiency among China's 31 provinces from 2010 to 2020 into three categories: high efficiency areas (0.876 to 0.915), medium efficiency areas (0.815 to 0.862), and low efficiency areas (0.7 to 0.77). The calculation outcomes are shown in Table 2, accurately characterizing regional differences.

K-means clustering partitioning and regional segmentation based on MHS efficiency reveal both similarities and significant differences. According to Table 2, within the high-efficiency range, only Beijing, Jiangsu, Shanghai, and Zhejiang, all originally from the eastern regions, remained in the high-efficiency range of MHS. Among them, Beijing, as the political, economic, and cultural center of the country, has a concentrating effect on high-quality medical and health resources. However, due to issues of "large investment, small output" and strict control over "medical treatment plus numbers," patients from other regions are unable to seek medical treatment in a timely manner, which has kept its MHS efficiency from being categorized in the high-efficiency zone. Conversely, Anhui, Henan, and Sichuan, located in the central and western areas, have moved into the highly efficient zones. This shift was primarily due to increased government policy support in these regions over recent years, which has significantly improved MHS efficiency. Most provinces in the central and western areas of China, such as Shanxi, Inner Mongolia, Jiangxi, Yunnan, and Guizhou, remained in the middle and

**Table 2. Cluster analysis of Chinese MHS efficiency between 2010 and 2020.**

| Efficiency range | Provincial regions |
|---|---|
| Low efficiency areas in medical services (0.7–0.77) | Tibet, Qinghai, Ningxia, Hainan |
| Medium efficiency areas in medical services (0.815–0.862) | Beijing, Heilongjiang, Jilin, Liaoning, Neimenggu, Shanxi, Shaanxi, Gansu, Xinjiang, Chongqing, Guizhou, Yunnan, Guangxi, Jiangxi, Fujian |
| High efficiency areas in medical services (0.876–0.915) | Sichuan, Hebei, Tianjin, Shandong, Jiangsu, Anhui, Shanghai, Zhejiang, Henan, Hubei, Hunan, Guangdong |

Table 3. Malmquist index and decomposition of Chinese MHS efficiency between 2010 and 2020.

| Age | Technical efficiency (EC) | Technnical advances (TC) | Pure Technical Efficiency (PC) | Scale efficiency (SC) | Total factor productivity (TFP) |
|---|---|---|---|---|---|
| 2010–2011 | 1.01 | 1.039 | 1.018 | 0.993 | 1.05 |
| 2011–2012 | 1.04 | 1.116 | 1.069 | 0.972 | 1.16 |
| 2012–2013 | 1.021 | 0.998 | 1.056 | 0.967 | 1.019 |
| 2013–2014 | 0.991 | 1.012 | 1.013 | 0.978 | 1.002 |
| 2014–2015 | 0.991 | 0.991 | 1.004 | 0.987 | 0.982 |
| 2015–2016 | 1.001 | 1.018 | 1.004 | 0.997 | 1.019 |
| 2016–2017 | 0.992 | 1.019 | 1.006 | 0.987 | 1.011 |
| 2017–2018 | 0.976 | 1.017 | 1.002 | 0.974 | 0.993 |
| 2018–2019 | 0.973 | 1.032 | 0.995 | 0.978 | 1.004 |
| 2019–2020 | 0.929 | 0.886 | 0.984 | 0.944 | 0.824 |
| 2010–2020 | 0.992 | 1.011 | 1.015 | 0.978 | 1.003 |

low efficiency areas, indicating a significant "Matthew effect" in the interprovincial MHS efficiency in China. Overall, MHS efficiency varied across different regions due to diversities in economic growth level, investment scale, and methods of medical resource allocation.

**3.1.2. Dynamic analysis based on Malmquist index.**   To thoroughly investigate the dynamic trends of Chinese MHS efficiency, the calculation outcomes of the Malmquist index from 2010 to 2020 are presented in Table 3 using DEAP2.1 software.

According to Table 3, the average TFPCH of Chinese MHS efficiency from 2010 to 2020 was greater than 1 (1.003), with an average annual increase of 0.3%, indicating an overall upward trend in national MHS efficiency and significant improvement. Notably, the TFPCH was highest from 2011 to 2012, at 1.16, signifying a 16% increase in total factor productivity. The TFPCH values for 2014–2015, 2017–2018, and 2019–2020 were all less than 1 (0.982, 0.993, and 0.824), indicating decreases in total factor productivity of 1.8%, 0.7%, and 17.6% respectively. From the perspective of decomposing indicators, the mean values of SC and EC were 0.978 and 0.992, respectively, meaning scale efficiency and pure technological efficiency decreased by 2.2% and 0.8% annually. Scale efficiency had a greater impact, highlighting that the insufficient total amount of medical and health resources still hindered the balanced development of the Chinese health industry. The average values of TC and PC were greater than 1 (1.011 and 1.015), indicating an increasing trend in both the technological progress index and the pure technical efficiency index, with the latter having the greatest impact. This analysis indicates that the primary direction for increasing the total factor productivity of Chinese MHS involves improving resource input and utility capabilities as the cornerstone, upgrading the input-output scale, medical equipment, and improving the skills and qualities of healthcare personnel, alongside the level of MHS technology and management.

## 3.2 Dynamic evolution of spatial model of MHS efficiency

**3.2.1. Spatial dependence analysis of MHS efficiency.**   ①*Global spatial autocorrelation analysis*. To demonstrate the spatial correlation of Chinese MHS efficiency, Stata17 software was used to analyze the overall Moran's I index from 2010 to 2020. Table 4 shows that the Moran's I values of Chinese MHS efficiency throughout the study period were all positive and statistically significant at the 1%, 5%, and 10% levels. This indicates that the spatial distribution of MHS efficiency across various provinces was not random, but exhibited a certain degree of spatial autocorrelation and clustering.

Furthermore, the Moran's I value did not decrease over time; instead, it exhibited an increasing trend, rising from 0.047 in the 2010–2012 period to 0.257 in the 2019–2020 period.

**Table 4. The Moran's index of whole autocorrelation for the efficiency of allocation between 2010 and 2020.**

| Age(year) | Moran's I | P |
|---|---|---|
| 2010–2012 | 0.047 | 0.001*** |
| 2013–2015 | 0.101 | 0.055* |
| 2016–2018 | 0.120 | 0.031*** |
| 2019–2020 | 0.257 | 0.001*** |

Note:

* is prominent at the 10% level, and * * * is prominent at the 1% level

This suggests that the spatial dependence of Chinese MHS efficiency did not show a significant weakening trend during the evaluation period, and the degree of spatial agglomeration increased annually. The reasons for this include each province actively integrating its own medical and health system status with national medical reform policies, exploring local medical reform paths, and reducing the correlation between provinces from 2010 to 2012, which coincided with the initial three-year period of the new Chinese medical reform. Subsequently, as healthcare reform progressed, various provinces and cities shared their experiences with each other, and the reform proceeded smoothly, gradually improving the spatial relevance of the Chinese healthcare industry.

②*Local spatial autocorrelation analysis*. The overall Moran's I index indicated a positive spatial correlation in MHS efficiency. To delve deeper into the local spatial correlation and agglomeration characteristics of each province, the Local Indicators of Spatial Association (LISA) were utilized to explore the spatial agglomeration characteristics of similar types of Chinese MHS efficiency.

As observed in Table 6, the LISA for the development of Chinese MHS efficiency from 2010 to 2020 displayed a "clustered" steady state with an increasingly clear trend of spatial homogeneity. Notably, the H-H clusters in Jiangsu, Zhejiang, Shanghai, and other regions in the east and central areas, as well as the L-L clusters in Qinghai, Xinjiang, Tibet, and other regions in the northwest were the most prominent. This indicated that the spatial structure of Chinese MHS efficiency was continuously evolving towards a "binary" structure, forming two types of convergence: high-level MHS efficiency convergence in the eastern and central areas and low-level MHS efficiency convergence in the western areas. Specifically, between 2010 and 2012, Chinese MHS efficiency exhibited an H-H clustering pattern primarily distributed in Shanghai, Zhejiang, Hunan, Hubei, Jiangxi, Fujian, and other areas, which had a relatively high level of self-development and were similar to their surrounding areas. The L-L agglomerations primarily appeared in Xinjiang, Tibet, Gansu, and other regions, which had a comparatively low level of development and significant differences from the surrounding areas. Between 2013 and 2015, H-H agglomerations were weakened due to Hunan and Zhejiang withdrawing from H-H clustering; L-L agglomerations had been strengthened because Qinghai evolved into an L-L cluster. From 2016 to 2018, H-H agglomerations were strengthened because Hunan and Zhejiang returned to H-H clustering; the L-L clustering provinces remained unchanged. During the period of 2019–2020, the L-L agglomerations were weakened because Gansu withdrew from the L-L cluster; the H-H clustering provinces remained unchanged. The evolution of Guizhou into an L-H cluster indicated that its MHS efficiency was significantly lower than that of surrounding areas, displaying a clear "center-periphery" characteristic and a certain polarization effect.

On the whole, Chinese MHS efficiency was mainly characterized by spatial homogeneity (H-H, L-L clustering), while spatial heterogeneity (L-H clustering) exhibited a certain sporadic

**Table 5. Elliptical Eigen values of the standard deviation of Chinese MHS efficiency between 2010 and 2020.**

| Elliptical area of age center coordinate /km² | Long axis standard deviation /km | Short axis standard deviation /km | Azimuth angle θ |
|---|---|---|---|
| 2010~2012 111.565˚E, 33.792˚N 3892986 | 118.422 | 104.646 | 54.565 |
| 2013~2015 111.585˚E, 33.791˚N 3900155 | 118.542 | 104.733 | 54.867 |
| 2016~2018 111.564˚E, 33.795˚N 3906819 | 118.653 | 104.813 | 55.146 |
| 2019~2020 113.052˚E, 33.594˚N 3912027 | 118.740 | 104.876 | 55.363 |

distribution. The H-H cluster was predominantly concentrated in the east and central areas such as Shanghai, Jiangsu, Anhui, Jiangxi, etc. These regions not only had developed economies and large investments in medical resources, advanced medical facilities, and technology but also hosted a large number of external workers, which somewhat increased the demand for medical services. Consequently, their MHS efficiencies were high and exerted a significant influence on the neighboring provinces. The L-L agglomeration was primarily located in western areas like Tibet and Qinghai, where the economic foundation was weaker, and medical resources and services were less accessible. It is noteworthy that in recent years, the emergence of L-H clusters, mainly in Guizhou and Hunan, has focused on developing their own MHS efficiency. However, the diffusion effect has not been significantly formed, while the polarization effect was increasing.

**3.2.2. Analysis of spatial evolution of MHS efficiency.** To interpret the directional deviation of the spatial distribution of Chinese MHS efficiency, we adopted the standard deviation ellipse method. This method enables a deep analysis of the spatial development features of Chinese MHS efficiency by observing the standard deviation ellipses of consecutive years. The calculation results are presented in Tables 5 and 6.

Results indicated that the center of Chinese MHS efficiency fluctuated between 111.564˚ to 113.052˚ E and 33.594˚ to 33.795˚ N from 2010 to 2020, and it was located in Henan Province, gradually moving towards the northwest direction. The interior of the ellipse covered the vast majorities of the southeastern coastal, central, and some western areas, which were the main players in the development of MHS efficiency. For example, MHS in Jiangsu and Shanghai maintained a high level; however, MHS in areas outside the ellipse, such as Tibet and Qinghai, remained at a relatively low level. Based on the LISA clustering, we also found that H-H

**Table 6. Dynamic evolution analysis of Chinese MHS efficiency between 2010 and 2020.**

| Agglomeration characteristics | Regions in 2010–2012 | Regions in 2013–2015 | Regions in 2016–2018 | Regions in 2019–2020 |
|---|---|---|---|---|
| not significant | Beijing,Helongjiang,Jilin,Liaoning, Neimenggu,Shanxi,Shaanxi, Sichuan,Chongqing,Yunnan, Guangxi,Guangdong,Hainan, Tianjin,Hebei,Shandong,Guizhou, Qinghai | Beijing, Helongjiang, Jilin, Liaoning, Neimenggu, Shanxi, Shaanxi, Sichuan, Chongqing, Yunnan, Guangxi, Guangdong, Hainan, Tianjin, Hebei, Shandong, Guizhou, Hunan, Zhejiang | Beijing, Helongjiang, Jilin, Liaoning, Neimenggu, Shanxi, Shaanxi, Sichuan, Chongqing, Yunnan, Guangxi, Guangdong, Hainan, Tianjin, Hebei, Shandong, Guizhou | Beijing, Helongjiang, Jilin, Liaoning, Neimenggu, Shanxi, Shaanxi, Sichuan, Chongqing, Yunnan, Guangxi, Guangdong, Hainan, Tianjin, Hebei, Shandong, Gansu, Ningxia |
| High-High Agglomeration | Henan,Hubei,Jiangsu,Anhui, Shanghai,Jiangxi,Fujian,Zhejiang, Hunan | Henan, Hubei, Jiangsu, Anhui, Shanghai, Jiangxi, Fujian | Henan, Hubei, Jiangsu, Anhui, Shanghai, Jiangxi, Fujian,Zhejiang, Hunan | Henan, Hubei,Jiangsu, Anhui, Shanghai, Jiangxi, Fujian,Zhejiang, Hunan |
| High-Low Agglomeration | - | - | - | - |
| Low-High Agglomeration | - | - | - | Guizhou |
| Low-Low Agglomeration | Xinjiang,Tibet,Gansu,Ningxia | Xinjiang, Tibet, Gansu, Ningxia, Qinghai | Xinjiang, Tibet, Gansu, Ningxia, Qinghai | Xinjiang, Tibet,Qinghai |

clustering was mostly located within the standard deviation ellipse, while L-L clustering was consistently located outside the standard deviation ellipse.

According to Table 5, the standard deviation ellipse exhibited an expanding trend in its distribution range. The elliptical area expanded from 3,892,986 km$^2$ during 2010–2012 to 3,912,027 km$^2$ during 2019–2020, and the elliptical shape gradually approached a circle. This indicated that the spatial distribution of Chinese MHS efficiency displayed an annual trend of diffusion. From the perspective of spatial distribution, the short axis standard deviation was consistently smaller than the long axis standard deviation, suggesting that Chinese MHS efficiency was mainly oriented in the northeast-southwest direction in its spatial distribution pattern. Specifically, both the long and short axes generally extended, with the long axis' standard deviation extending from 118.422 km during 2010–2012 to 118.74 km during 2019–2020, and the short axis' standard deviation extending from 104.646 km during 2010–2012 to 104.876 km during 2019–2020. Although the overall changes were relatively small, they indicated that the spatial distribution of Chinese MHS efficiency followed a scattered trend in the "northeast-southwest" direction. Regarding the azimuth angle θ, the turning angle expanded from 54.565˚ in 2010 to 55.363˚ in 2020, suggesting that the spatial distribution pattern of MHS efficiency rotated clockwise from northeast-southwest to a more defined east-west orientation by 0.798˚, thus strengthening the northeast-southwest pattern.

## 4. Discussion

Firstly, from the static analysis using the SFA model, we observed that the average MHS efficiency in China between 2010 and 2020 was 0.845. While this overall efficiency level was not high, it demonstrated that Chinese MHS efficiency was continuously improving and maintained a long-term positive trend. According to the dynamic analysis results of the Malmquist index, the average TFPCH of MHS in China during the research period was 1.003, with an average annual improvement of 0.3%. The principal drivers of this increase were improvements in pure technological efficiency, which had a more significant impact than scale efficiency. Therefore, it is essential to strengthen the construction of high-tech medical equipment, improve the training and development of professional talents, and optimize the "soft power" of medical institutions to further improve total factor productivity [44, 45]. From the K-means clustering analysis results, Chinese MHS efficiency exhibited regional differentiation, characterized as "high in the eastern and central areas, low in the northwestern regions." There were significant diversities in MHS efficiency among different provinces within their original geographical areas. The efficiency levels in eastern and central areas such as Shanghai, Jiangsu, Guangdong, and Anhui have consistently been in the high-efficiency zones for many years, while the northeastern and western regions, represented by Jilin, Heilongjiang, Inner Mongolia, Yunnan, and Guizhou, have remained in the medium and low efficiency zones, demonstrating a significant gap with the eastern and central areas. This disparity indicates a substantial Matthew effect in Chinese MHS efficiency among different provinces, becoming an urgent issue for the sustainable growth of the Chinese medical and health industry. Looking forward, medical and health administrative departments should continue to deeply implement the regional public health coordinated development strategy, continuously promote balanced development between regions, and share high-quality medical and health resources [46, 47].

Secondly, the analysis of overall spatial autocorrelation revealed a significant spatial positive dependence in the distribution of MHS efficiency among provinces in China from 2010 to 2020. The geographical spatial distribution demonstrated a significant agglomeration effect between provinces. In this context, various provinces should improve their own MHS efficiency as well as monitor and adapt to the operational and evolutionary trends of

interprovincial medical and health systems in surrounding areas to mitigate their potential negative spatial impacts. According to the results of the local spatial autocorrelation study, the efficiency clustering characteristics of Chinese MHS during the research period predominantly consisted of H-H and L-L types of clustering, displaying a continuous regional distribution pattern. The H-H cluster primarily included Hubei, Anhui, and Zhejiang, while the L-L cluster mainly comprised Tibet and Qinghai. The L-H clustering area that emerged in 2019 was exclusive to Guizhou Province, which correlated with its low efficiency value. There was no H-L aggregation observed. Therefore, provinces within the H-H clustering should assume a leading and exemplary role, facilitating the cross-provincial flow of professional health talents, technology, and other resource elements, thereby promoting the effective improvement of interprovincial MHS efficiency in surrounding areas. Provinces within L-L and L-H clustering should adopt best practices from more efficient provinces, improve their organizational management, and other construction aspects to improve the continuous MHS efficiency of their own. The spatial dynamic evolution characteristics of Chinese MHS efficiency during the study period were as follows: the center of MHS efficiency fluctuated between 111.564° to 113.052° E and 33.594° to 33.795° N, consistently located within the territory of Henan and moving in a northwest direction overall. The standard deviation ellipse indicated that MHS efficiency followed a northeast-southwest pattern with a continuous trend of diffusion towards the northwest. Compared to other provinces, the level of interprovincial MHS in the western regions remained relatively low, with slow development and less influence from the radiation of surrounding areas. In the future, efforts should continue to utilize the eastern and central regions as entry points, radiating and driving surrounding areas and forming a high-level MHS development axis with horizontal expansion from east to central to northwest.

## 5. Conclusion

This article studied panel data from 31 Chinese provinces and regions between 2010 and 2020, employing the SFA Malmquist model to explore the dynamic and static trends of MHS efficiency. Additionally, methods like spatial correlation analysis and the standard deviation ellipse were utilized to reveal the spatial clustering characteristics and development processes of interprovincial MHS efficiency. The primary conclusions are as follows:

Firstly, regarding efficiency values, throughout the research timeframe, the overall level of MHS efficiency across the 31 Chinese provinces was not high, displaying a weak growth trend. There was a notable differentiation feature of "high in the east and in the middle, low in the northwest," with interprovincial differences becoming more pronounced. In terms of total factor productivity, Chinese total MHS factor productivity showed an upward trend from 2010 to 2020, with PC and TC playing leading roles in TFP improvement. Secondly, under spatial correlation analysis, a positive spatial correlation of MHS efficiency among the 31 provinces of China was observed, and the phenomenon of spatial agglomeration increased year by year. Spatial clustering was primarily characterized by H-H and L-L clustering, with L-H clustering appearing sporadically. Specifically, H-H clustering was predominantly found in the eastern and central areas of Anhui and Jiangsu. L-H clustering was located in the central and western areas of Guizhou and Hunan, while L-L clustering was primarily in the western regions of Tibet and Qinghai, with the former having a greater number than the latter, positively influencing overall Chinese MHS efficiency. According to the standard deviation ellipse analysis, Chinese MHS efficiency followed a spatial model extending from northeastern to southwestern areas, with its center consistently located within the territory of Henan Province.

In summary, this article proposed the following suggestions: (1) There should be significant attention paid to the equal development of MHS. Increasing investment in medical and health

resources and their allocation efficiency in the northeast and west is essential to gradually narrow the gap in MHS and upgrade harmonized regional growth. (2) It is crucial to focus on strengthening technological innovation and cultivating high-quality health talents. This involves upgrading the level of professional and technical advancement, improving the internal management level and capability of medical and health agencies, and strengthening the refined management of the MHS system to significantly improve its operational efficiency. (3) The government should fully engage with the spatial positive autocorrelation and spatial clustering phenomenon of Chinese MHS efficiency. Playing a leading role in H-H regions like Shanghai, Jiangsu, Anhui, and Jiangxi, and accelerating the orderly flow of high-quality medical and health resources to L-L regions like Tibet and Qinghai and L-H regions like Guizhou and Hunan is crucial to avoid the spread of the Matthew effect. Additionally, integrating fragmented medical resources and establishing a mechanism for the flow of medical resources between regions will promote regional linkage, make it easier for people to access remote MHS, and upgrade the level of service provided.

## Author Contributions

**Data curation:** Jing Zhang.

**Formal analysis:** Jing Zhang.

**Project administration:** Jing Zhang.

**Validation:** Jing Zhang.

**Visualization:** Jing Zhang.

**Writing – original draft:** Jing Zhang.

**Writing – review & editing:** Jing Zhang.

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
