## [Decision Letter · Decision Letter 0]

26 Mar 2024

PONE-D-24-05935Research on the Dynamic Evolution of Utilization Efficiency of Medical and Health Services in ChinaPLOS ONE

Dear Dr. ZHANG,

Thank you for submitting your manuscript to PLOS ONE. After careful consideration, we feel that it has merit but does not fully meet PLOS ONE’s publication criteria as it currently stands. Therefore, we invite you to submit a revised version of the manuscript that addresses the points raised during the review process.

**ACADEMIC EDITOR: **The phrase "Research on” should be removed from the title. 

Revise and refine the language in the manuscript.

We look forward to receiving your revised manuscript.

Kind regards,

Xufeng Cui, Ph.D

Academic Editor

PLOS ONE

Journal Requirements:

2. Thank you for submitting the above manuscript to PLOS ONE. During our internal evaluation of the manuscript, we found significant text overlap between your submission and previous work in the [introduction, conclusion, etc.].

Please revise the manuscript to rephrase the duplicated text, cite your sources, and provide details as to how the current manuscript advances on previous work. Please note that further consideration is dependent on the submission of a manuscript that addresses these concerns about the overlap in text with published work.

[If the overlap is with the authors’ own works: Moreover, upon submission, authors must confirm that the manuscript, or any related manuscript, is not currently under consideration or accepted elsewhere. If related work has been submitted to PLOS ONE or elsewhere, authors must include a copy with the submitted article. Reviewers will be asked to comment on the overlap between related submissions (http://journals.plos.org/plosone/s/submission-guidelines#loc-related-manuscripts).]

We will carefully review your manuscript upon resubmission and further consideration of the manuscript is dependent on the text overlap being addressed in full. Please ensure that your revision is thorough as failure to address the concerns to our satisfaction may result in your submission not being considered further.

"This work was supported by the Major project of Humanities and Social Sciences "Entrusted Production License and Supervision Strategy of Traditional Chinese Medicine Decoction Pieces" (No.: SK2020ZD22)."

"The authors received no specific funding for this work."

5. Please provide a complete Data Availability Statement in the submission form, ensuring you include all necessary access information or a reason for why you are unable to make your data freely accessible. If your research concerns only data provided within your submission, please write "All data are in the manuscript and/or supporting information files" as your Data Availability Statement.

Additional Editor Comments:

The phrase "Research on” should be removed from the title.

Revise and refine the language in the manuscript.

Reviewers' comments:

Reviewer's Responses to Questions

**Comments to the Author**

1. Is the manuscript technically sound, and do the data support the conclusions?

Reviewer #1: Partly

Reviewer #2: Yes

2. Has the statistical analysis been performed appropriately and rigorously? 

Reviewer #1: I Don't Know

Reviewer #2: Yes

3. Have the authors made all data underlying the findings in their manuscript fully available?

Reviewer #1: Yes

Reviewer #2: Yes

4. Is the manuscript presented in an intelligible fashion and written in standard English?

Reviewer #1: Yes

Reviewer #2: Yes

5. Review Comments to the Author

Reviewer #1: The paper uses the SFA Malmquist model, the spatial econometric model, and the standard deviation ellipse method to evaluate the efficiency, productivity, spatial dependence, and dynamic evolution of medical and health services in 31 provinces of China, based on data from statistical yearbooks.

The authors declaim that the average level of efficiency and productivity of medical and health services in China is moderate and has a weak growth trend, with significant regional differences and spatial clustering. The paper also identifies the factors that affect the efficiency and productivity, such as population density, government health budget, and technological innovation.

However, this paper has some limitations, such as:

(1) The paper does not consider the quality and accessibility of medical and health services, which are important aspects of health system performance.

(2) The paper does not account for the impact of the COVID-19 pandemic on the medical and health services in 2020, which may have distorted the results and trends.

(3) The paper does not provide any policy recommendations or implications based on the findings, which may limit its practical relevance and applicability.

(4) The paper uses the SFA model and the Malmquist index to measure the MHS efficiency, but does not address the potential limitations or biases of these methods, such as the choice of input and output indicators, the specification of the production function, the assumption of homogeneity, etc.

Reviewer #2: The study titled “Research on the Dynamic Evolution of Utilization Efficiency of Medical and Health Services in China focuses on evaluating the efficiency of the Chinese medical and health system. The research investigates an important topic. However, it needs extensive revision to improve its quality. My details comments are as below.

• The abstract should present the research objectives, methods, and concise results of the study. Revise these concerns.

• The innovations, objectives, and contributions are missing from the introduction section. Revise it and write the importance of the topic, problem, objectives, contributions, and innovations.

• Rewrite the following paragraph in the introduction section for more clearance” Domestic scholars have conducted extensive researches on the medical services efficiency. Based on medical and health data of Chinese 30 provinces from 2005 to 2018, Li Yi (2021)[10] explored the MHS efficiency and its interfering factors. The results showed that the MHS efficiency had a strong trend in the East, and weak in the West and Central. The developing level of regional economy and technology education had a positive effect on medical and health, while regional aging and transportation levels would affect MHS efficiency. Zhang Mei (2022)[11] evaluated the public MHS efficiency in traditional Chinese medicine hospitals from 2012 to 2020 and found that the overall medical services efficiency in Chinese traditional medicine hospitals was at a moderate level with significant regional differences. The external environment, such as the average times and sickbeds per day that Chinese medicine hospital physicians were responsible for treatment and diagnosis, significantly affected the MHS efficiency. Luan Yunyun (2022)[12] analyzed the interfering factors and regional efficiency differences of MHS efficiency since the " China New Medical Reform", and found that the differences of medical services efficiency in the eastern, central and western regions of China were showing a downward trend. Population density, government health budget expenditures, and the non-market level of medical services significantly affected the medical services efficiency. The above research indicated that the medical services efficiency was not only related to the type of medical institution and organizational management but also significantly influenced by exogenous variables such as population density.

• Add the following latest literature on the topic of efficiency estimation. https://doi.org/10.1016/j.energy.2022.124507

https://doi.org/10.1007/s11356-022-23484-w

https://doi.org/10.1016/j.egyr.2022.12.067

https://doi.org/10.1016/j.gr.2023.07.017

https://doi.org/10.1016/j.gsf.2023.101631

https://doi.org/10.3390/f15010152

https://doi.org/10.1016/j.scitotenv.2023.168027

https://doi.org/10.1016/j.egyr.2022.07.161

https://doi.org/10.1007/s11356-022-19729-3

• Check the table numbers.

• In the methodology section add the importance of the Malmquist productivity index.

• The selection of inputs and outputs has great significance in efficiency estimation as it could impact the results. Mention the reference from where the authors choose these variables.

• Results should be backed by literature. Add more discussion and add citations to back your results.

• Add concise and comprehensive study results and more policy recommendations in the conclusion section for the medical sector of China.

• Avoid grammatical errors throughout the manuscript.

• Referencing style is inappropriate. Use proper citation style according to the Plos One requirement.

6. PLOS authors have the option to publish the peer review history of their article (what does this mean?). If published, this will include your full peer review and any attached files.

Reviewer #1: No

Reviewer #2: No

---

## [Author Response · Author response to Decision Letter 0]

20 Apr 2024

Response to Reviewers

Dear Academic Editor:

Thank you for giving me the opportunity to submit a revised draft of the manuscript “the Dynamic Evolution of Utilization Efficiency of Medical and Health Services in China” for publication in the Journal of PLOS ONE. I appreciate the time and effort that you and the reviewers dedicated to providing feedback on our manuscript and are grateful for the insightful comments on and valuable improvements to my paper. I have incorporated most of the suggestions made by you and the reviewers. Those changes are highlighted in the manuscript. Please see answers below, for a point-by-point response to you and the reviewers’ comments and concerns. All page numbers refer to the revised manuscript file with tracked changes.

Additional Editor Comments:

The phrase "Research on” should be removed from the title.

Revise and refine the language in the manuscript.

Answers: The phrase "Research on” had been removed from the title. The manuscript had also been revised and the language refined.

Reviewer #1: The paper uses the SFA Malmquist model, the spatial econometric model, and the standard deviation ellipse method to evaluate the efficiency, productivity, spatial dependence, and dynamic evolution of medical and health services in 31 provinces of China, based on data from statistical yearbooks.

The authors declaim that the average level of efficiency and productivity of medical and health services in China is moderate and has a weak growth trend, with significant regional differences and spatial clustering. The paper also identifies the factors that affect the efficiency and productivity, such as population density, government health budget, and technological innovation.

However, this paper has some limitations, such as:

(1) The paper does not consider the quality and accessibility of medical and health services, which are important aspects of health system performance.

Answers: The data of the quality and accessibility of medical and health services are difficult to get, so indexes of the quality and accessibility of medical and health services were not included. The article chose other indexes like input-output to estimate.

(2) The paper does not account for the impact of the COVID-19 pandemic on the medical and health services in 2020, which may have distorted the results and trends.

Answers: The data used in the article were sourced from the 2010-2010 “China Statistical Yearbook”, “ China Health Statistical Yearbook” and“China Health and Family Planning Statistical Yearbook”. The data of the COVID-19 pandemic on the medical and health services were included in these yearbooks. 

(3) The paper does not provide any policy recommendations or implications based on the findings, which may limit its practical relevance and applicability.

Answers: Added in the last paragraph of the whole article.

(4) The paper uses the SFA model and the Malmquist index to measure the MHS efficiency, but does not address the potential limitations or biases of these methods, such as the choice of input and output indicators, the specification of the production function, the assumption of homogeneity, etc.

Answers: Literature survey revealed that the SFA model and the Malmquist index are a specialized index analysis method for measuring the efficiency, they are widely used in the calculation of production efficiency in sectors such as finance, industry, and healthcare. So the potential limitations or biases of them are ignored in the article.

Reviewer #2: The study titled “Research on the Dynamic Evolution of Utilization Efficiency of Medical and Health Services in China focuses on evaluating the efficiency of the Chinese medical and health system. The research investigates an important topic. However, it needs extensive revision to improve its quality. My details comments are as below.

• The abstract should present the research objectives, methods, and concise results of the study. Revise these concerns.

Answers: These concerns have been done.

• The innovations, objectives, and contributions are missing from the introduction section. Revise it and write the importance of the topic, problem, objectives, contributions, and innovations.

Answers: (1)The first three sentences in “Introduction” have revealed the importance of the topic. (2)The second paragraph beginning with “However” in “Introduction” have revealed the current problems in the developing of MHS, and the fifth paragraph beginning with “However” in “Introduction” have revealed the current problems in the research field of MHS. (3)The last sentences of Paragraph 2 in “Introduction” were the objectives.(4)The last paragraph has been written for the contributions of this article.(5)The innovations were expressed in paragraph 6 in “Introduction”, some sentences were modified and the innovations could be clear.

• Rewrite the following paragraph in the introduction section for more clearance” Domestic scholars have conducted extensive researches on the medical services efficiency. Based on medical and health data of Chinese 30 provinces from 2005 to 2018, Li Yi (2021)[10] explored the MHS efficiency and its interfering factors. The results showed that the MHS efficiency had a strong trend in the East, and weak in the West and Central. The developing level of regional economy and technology education had a positive effect on medical and health, while regional aging and transportation levels would affect MHS efficiency. Zhang Mei (2022)[11] evaluated the public MHS efficiency in traditional Chinese medicine hospitals from 2012 to 2020 and found that the overall medical services efficiency in Chinese traditional medicine hospitals was at a moderate level with significant regional differences. The external environment, such as the average times and sickbeds per day that Chinese medicine hospital physicians were responsible for treatment and diagnosis, significantly affected the MHS efficiency. Luan Yunyun (2022)[12] analyzed the interfering factors and regional efficiency differences of MHS efficiency since the " China New Medical Reform", and found that the differences of medical services efficiency in the eastern, central and western regions of China were showing a downward trend. Population density, government health budget expenditures, and the non-market level of medical services significantly affected the medical services efficiency. The above research indicated that the medical services efficiency was not only related to the type of medical institution and organizational management but also significantly influenced by exogenous variables such as population density.

Answers: The paragraph had been revised.

• Add the following latest literature on the topic of efficiency estimation. https://doi.org/10.1016/j.energy.2022.124507

https://doi.org/10.1007/s11356-022-23484-w

https://doi.org/10.1016/j.egyr.2022.12.067

https://doi.org/10.1016/j.gr.2023.07.017

https://doi.org/10.1016/j.gsf.2023.101631

https://doi.org/10.3390/f15010152

https://doi.org/10.1016/j.scitotenv.2023.168027

https://doi.org/10.1016/j.egyr.2022.07.161

https://doi.org/10.1007/s11356-022-19729-3

Answers: The above latest literature had been added.

• Check the table numbers.

Answers: Checked.

• In the methodology section add the importance of the Malmquist productivity index.

Answers: Added.

• The selection of inputs and outputs has great significance in efficiency estimation as it could impact the results. Mention the reference from where the authors choose these variables.

Answers: The input and output data of medical resources in Chinese 31 provinces were sourced from the 2010-2010 “China Statistical Yearbook”, “ China Health Statistical Yearbook” and“China Health and Family Planning Statistical Yearbook”. This sentence has been expressed in the article with the refence[45].

• Results should be backed by literature. Add more discussion and add citations to back your results.

Answers: Added.

• Add concise and comprehensive study results and more policy recommendations in the conclusion section for the medical sector of China.

Answers: Added in the last paragraph of the whole article.

• Avoid grammatical errors throughout the manuscript.

Answers: I have revised as possible as I can.

• Referencing style is inappropriate. Use proper citation style according to the Plos One requirement.

Answers: Referencing style had been revised according to the Plos One requirement.

Kind regards,

Jing ZHANG, Ph.D

---

## [Decision Letter · Decision Letter 1]

8 May 2024

Dynamic Evolution of Utilization Efficiency of Medical and Health Services in China

PONE-D-24-05935R1

Dear Dr. ZHANG,

We’re pleased to inform you that your manuscript has been judged scientifically suitable for publication and will be formally accepted for publication once it meets all outstanding technical requirements.

Kind regards,

Xufeng Cui, Ph.D

Academic Editor

PLOS ONE

Additional Editor Comments (optional):

Reviewers' comments:

Reviewer's Responses to Questions

**Comments to the Author**

1. If the authors have adequately addressed your comments raised in a previous round of review and you feel that this manuscript is now acceptable for publication, you may indicate that here to bypass the “Comments to the Author” section, enter your conflict of interest statement in the “Confidential to Editor” section, and submit your "Accept" recommendation.

Reviewer #1: All comments have been addressed

Reviewer #2: All comments have been addressed

2. Is the manuscript technically sound, and do the data support the conclusions?

Reviewer #1: (No Response)

Reviewer #2: Yes

3. Has the statistical analysis been performed appropriately and rigorously? 

Reviewer #1: I Don't Know

Reviewer #2: Yes

4. Have the authors made all data underlying the findings in their manuscript fully available?

Reviewer #1: Yes

Reviewer #2: Yes

5. Is the manuscript presented in an intelligible fashion and written in standard English?

Reviewer #1: No

Reviewer #2: Yes

6. Review Comments to the Author

Reviewer #1: (No Response)

Reviewer #2: I can see that Authors has carefully revised the manuscript. no more revision required. Therefore, I recommend acceptance in the current form.

7. PLOS authors have the option to publish the peer review history of their article (what does this mean?). If published, this will include your full peer review and any attached files.

Reviewer #1: No

Reviewer #2: No

---

## [Editor Report · Acceptance letter]

14 Jun 2024

PONE-D-24-05935R1 

PLOS ONE

Dear Dr. Zhang, 

I'm pleased to inform you that your manuscript has been deemed suitable for publication in PLOS ONE. Congratulations! Your manuscript is now being handed over to our production team.

Kind regards, 

on behalf of

Professor Xufeng Cui 

Academic Editor

PLOS ONE